# Cultivation of Vitamin C-Rich Vegetables for Space-Radiation Mitigation

**Alireza Mortazavi** [1,†], **Helia Yarbaksh** [2,†], **Batool Faegheh Bahaaddini Baigy Zarandi** [3], **Reza Yarbakhsh** [4], **Fatemeh Ghadimi-Moghaddam** [5], **Syed Mohammad Javad Mortazavi** [6], **Masoud Haghani** [7], **Donya Firoozi** [2,8] and **Lembit Sihver** [9,10,11,*]

1    MVLS College, The University of Glasgow, Glasgow G12 8QQ, UK; 2921617m@student.gla.ac.uk
2    Student Research Committee, School of Nutrition and Food Sciences, Shiraz University of Medical Sciences, Larestan 7433193514, Iran; h.yarbakhsh@larums.ac.ir (H.Y.); donyafiroozi@gmail.com (D.F.)
3    Department of Pharmacology, School of Medicine, Shiraz University of Medical Sciences, Shiraz 7134845794, Iran; zarandi@sums.ac.ir
4    Department of Computer Engineering, Sharif University of Technology, Tehran 1411713114, Iran; ryarbakhsh@gmail.com
5    Ionizing and Non-Ionizing Radiation Protection Research Center (INIRPRC), Albert Szent-Györgyi Medical University, H-6720 Szeged, Hungary; ghadimimoghadam.fatemeh@o365.u-szeged.hu
6    Ionizing and Non-Ionizing Radiation Protection Research Center (INIRPRC), Shiraz University of Medical Sciences, Shiraz 7143918596, Iran; mmortazavi@sums.ac.ir
7    Department of Radiology, School of Paramedical Sciences, Shiraz University of Medical Sciences, Shiraz 7143918596, Iran; m.haghani4744@yahoo.com
8    Department of Decision Sciences, University of Montreal, Montreal, QC H3T 2A7, Canada
9    Department of Radiation Physics, Atominstitut, Technische Universität, 1020 Vienna, Austria
10   Department of Physics, East Carolina University, Greenville, NC 27858, USA
11   Department of Chemistry and Chemical Engineering, Royal Military College of Canada, Kingston, ON K7K 7B4, Canada
*    Correspondence: lembit.sihver@tuwien.ac.at; Tel.: +46-721-726431
†    These authors contributed equally to this work.

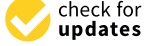



**Simple Summary:** Space exploration introduces astronauts to many challenges, such as space radiation and microgravity. We have investigated the use of vitamin C as a potential radiation mitigator, and the use of antioxidants in sustaining astronaut health. The cultivation of vitamin C-rich vegetables, which astronauts can consume after the being exposed to sudden large solar particle events, could help mitigate acute radiation illness.

**Abstract:** Space exploration introduces astronauts to challenges, such as space radiation and microgravity. Researchers have investigated vitamin C as a potential radiation mitigator, as well as antioxidants for sustaining astronaut health. Our own studies demonstrate vitamin C's life-saving radioprotective effects and its potential as a radiation mitigator, thus highlighting promise, even when administered 24 h post-exposure. This is particularly relevant in scenarios where astronauts may be exposed to sudden large solar particle events, potentially resulting in lethal doses of space radiation. The success of vegetable cultivation on the International Space Station using NASA's Veggie system offers fresh, vitamin C-rich food. While approved supplements address somatic function, further research is needed to optimize vitamin C's efficacy in humans, and to develop appropriate antioxidant cocktails for space missions. The variable vitamin C content in vegetables underscores the necessity for the utilization of artificial intelligence (AI) to assist astronauts in selecting and cultivating the vitamin C-rich vegetables best-suited to combat high levels of space radiation and microgravity. Particularly, AI algorithms can be utilized to analyze various factors, such as nutritional content, growth patterns, and cultivation methods. In conclusion, vitamin C shows significant potential for mitigating space radiation, and ongoing research aims to enhance astronaut health through optimal dietary strategies.

**Keywords:** radiation; C-vitamin; solar particle event; SPE; antioxidants; radiation mitigator

## 1. Introduction

Space exploration poses unique challenges to astronauts, including exposure to space radiation and microgravity [1]. To mitigate the detrimental effects of these stressors, researchers have investigated the potential of vitamin C as a radiation mitigator and have explored the use of antioxidants in maintaining astronaut health [2]. Studies on rodents have shown that vitamin C can reduce radiation-induced chromosomal damage. Additionally, experiments exposing rates to gamma radiation have demonstrated improved survival rates and cell viability when treated with vitamin C [3]. NASA has successfully tested vegetable cultivation aboard the International Space Station (ISS) using the Veggie system, providing fresh food and dietary variety for astronauts [4]. While vegetables which are high in vitamin C offer health benefits, limited research exists on their efficacy in coping with space radiation and microgravity. Moreover, the vitamin C content of these vegetables can vary, highlighting the need for further investigation. Supplements regulating somatic functions have been approved for reducing the effects of space radiation, microgravity, and the hypo-magnetic environment in space [5]. Our team's studies have indicated that vitamin C has emerged as a prospective radiation mitigator, even when administered 24 h after radiation exposure [2]. This is particularly relevant in scenarios where astronauts may be exposed to sudden large solar particle events, potentially resulting in lethal doses of space radiation. Furthermore, the development of an antioxidant cocktail is being considered as a strategy to counteract the damage caused by reactive oxygen species (ROS) during space exploration [6]. In conclusion, vitamin C shows promise as a radiation mitigator, while antioxidants hold potential for protecting against space-induced stressors.

Natural antioxidants are widely distributed in food and medicinal plants, such as fruits, vegetables, cereals, mushrooms, beverages, flowers, spices, and traditional medicinal herbs. These natural antioxidants from plant materials are mainly polyphenols (phenolic acids, flavonoids, anthocyanins, lignans, and stilbenes), carotenoids (xanthophylls and carotenes), and vitamins (vitamin E and C) [7]. Besides vitamin C, other antioxidant compounds in plants might offer a broader spectrum of antioxidant properties that are crucial in mitigating radiation effects. Plants such as Podophyllum hexandrum, Amaranthus paniculatus, Emblica officinalis, Phyllanthus amarus, Piper longum, Tinospora cordifolia, Mentha arvensis, Mentha piperita, Syzygium cumini, Zingiber officinale, Ageratum conyzoides, Aegle marmelos, and Aphanamixis polystachya have been shown to protect against radiation-induced lethality, lipid peroxidation, and DNA damage [8].Further research is warranted to determine the effectiveness of vitamin C in humans, and to optimize antioxidant formulations for maintaining astronaut health during space missions.

## 2. Harsh Space Environment

The space environment is a formidable and challenging realm, characterized by a multitude of stressors that can have profound effects on the human body. Among these stressors, radiation and microgravity stand out as significant contributors to physiological alterations and potential interplays [9,10]. The unforgiving nature of space poses a myriad of challenges for human space exploration, with environmental stressors playing a pivotal role in shaping the physiological responses of astronauts. This paper will delve into the intricacies of two prominent stressors—radiation and microgravity—and examine their unique characteristics [11], individual effects, and the potential interplays that may amplify their impact on the human body. In this section, we seek to provide a comprehensive overview of the harsh space environment, with a particular emphasis on radiation and microgravity, exploring their individual impacts and potential synergistic effects on human health.

## 3. Space Radiation

The primary threat to the health of astronauts on long-duration space exploration missions outside of Earth's magnetosphere is space radiation [12]. Space is permeated by various forms of radiation, including cosmic rays, solar radiation, and trapped particles in planetary magnetic fields [13]. The high-energy nature of these radiations poses a significant threat to astronauts, as ionizing radiation can penetrate spacecraft shielding and interact with biological tissues [12]. The biological effects of radiation exposure include DNA damage, cellular mutations, and an increased risk of cancer. Developing effective radiation shielding and mitigation strategies remains a critical aspect of space exploration endeavors.

Numerous studies have shown that high-energy nuclei, such as HZE particles, have a greater relative biological effectiveness (RBE) in causing cancer-related changes when compared to low-LET radiation [14]. Experiments conducted in test tubes and outside the living organism have demonstrated that HZE particles like 56Fe, C, and He can lead to significant abnormalities in chromosomes and mutations in various types of cells. These findings align with the increased potential associated with HZE particles in combatting cancer, as observed in studies on animals. Results from experiments conducted within living organisms support the findings from test tube studies, and confirm the significantly greater ability of HZE particles to initiate cancer-related changes when compared to low-LET radiation [14].

Space radiation is a major concern for deep space missions, due to its potential to cause cognitive alterations in astronauts. In a recent study on Wistar rats, the NK1 receptor antagonist rolapitant and the anxiolytic diazepam were tested to understand their effects on IR-induced cognitive alterations. The study found that rolapitant did not significantly change anxiety, locomotor activity, or cognitive abilities in the treated rats under irradiation. However, it did affect the protein amount of certain serotonin receptors in the amygdala of irradiated rats. The study suggests that rolapitant may be a potential atypical radioprotector for treating central nervous system functional disorders caused by IR in astronauts [15].

## 4. Vitamin C in Radiotherapy

Vitamin C has been studied for its potential to mitigate the effects of radiation in space medicine and enhance the effects of radiation therapy in terrestrial medicine. In space, vitamin C has been proposed as a potential non-toxic and cheap radiation mitigator for astronauts on deep space missions, as it can help reduce radiation-induced health effects and chromosomal damage, and act as an effective antioxidant and free radical scavenger. On Earth, the effects of vitamin C in combination with radiation therapy have been studied in the context of cancer treatment. While some in vitro studies have shown radio-sensitizing effects of pharmacological doses of vitamin C combined with irradiation in certain cancer types, the impact of combining vitamin C with radiation therapy in human breast cancer has yielded varying results, with lower doses of vitamin C potentially increasing cancer cell proliferation and decreasing radiosensitivity. Therefore, the role of vitamin C as a radiation mitigator in space medicine and a radiation radiosensitizer in terrestrial medicine is complex and context-dependent, and further research is needed to fully understand its effects in both settings.

Khazaei et al. have shown that the combination of vitamin C with radiotherapy can lead to a non-significant increase in colony formation, as well as minor differences in cell cycle arrest and protein expression when compared to radiotherapy (RT) alone [16]. Lower doses of vitamin C, either alone or combined with radiotherapy, led to higher cell proliferation in a manner specific to each cell line. The use of vitamin C was linked to a lower histological grade and BMI, but not to an increased risk of recurrence in RT-treated patients. The impact of vitamin C on RT efficiency varied depending on the subtype of breast cancer and the concentration of vitamin C. Lower doses of vitamin C, which can be achieved through oral administration, may lead to an increased proliferation of breast cancer cells and a reduced sensitivity to radiation.

## 5. Microgravity

Microgravity, or the partial absence of the Earth's gravitational forces as experienced in ISS, leads to unique physiological adaptations in the human body. Skeletal muscle atrophy [17,18], bone demineralization [19,20], and alterations in cardiovascular function [21,22] are among the well-documented consequences of prolonged exposure to microgravity. The absence of gravitational cues challenges the body's homeostatic mechanisms, necessitating countermeasures to mitigate the deleterious effects on astronaut health.

Research has demonstrated that microgravity can impact the growth and nutritional value of wheatgrass, a plant known for its high levels of antioxidants like vitamin C. Simulated microgravity conditions have been found to increase wheatgrass's antioxidant activity, the total phenolic content, the total flavonoid content, and its effects on breast cancer cell lines [23]. Wheatgrass grown under microgravity conditions has shown higher levels of vitamin C, which may contribute to its improved antioxidant properties [24]. However, there is currently no direct evidence on how microgravity affects the role of vitamin C as a radiation mitigator in space medicine or as a radiosensitizer in terrestrial medicine.

In terrestrial medicine, vitamin C has been researched for its potential to enhance the effects of radiation therapy in cancer treatment. However, the results are mixed, with some studies indicating radio-sensitizing effects, while others show no significant benefit or even a decrease in radiosensitivity [24]. In space medicine, vitamin C has been suggested as a radiation mitigator, but its role in this context has not been extensively studied.

## 6. Potential Interplays

While radiation and microgravity represent distinct stressors, their potential interplays in space require careful consideration [11]. Studies suggest that microgravity may exacerbate the effects of radiation exposure on bone density and muscle mass. The compromised immune function observed in microgravity could further amplify the susceptibility of astronauts to radiation-induced health risks. Understanding these potential interplays is crucial for developing comprehensive countermeasures that address the combined impact of these stressors during extended space missions.

Given these considerations, the space environment poses numerous challenges to human health, with radiation and microgravity standing out as key stressors [11]. As space exploration ventures extend to more prolonged missions, the cumulative effects of these stressors and their potential interplays demand thorough investigation. Developing effective countermeasures and spacecraft design strategies to mitigate the harsh space environment's impact on human physiology is imperative for the success of future space exploration missions. As we venture further into the cosmos, the scientific community must continue to refine our understanding of these stressors, working collaboratively to safeguard the well-being of astronauts during their journeys beyond Earth.

## 7. Physiological Implications of Prolonged Space Habitation

The formidable threat posed by space radiation, composed of high-energy particles emanating from the sun and cosmic sources, as well as microgravity, manifests in DNA damage and compromised immune system functionality. This elevation in the susceptibility of astronauts to various illnesses [25–27] underscores the critical imperative of identifying and implementing effective countermeasures to mitigate the adverse impacts of these stressors, ensuring the success and well-being of astronauts during extended space missions.

The physiological ramifications of extended space habitation have become a focal point of inquiry as human exploration ventures beyond the confines of Earth. Among the myriad of stressors encountered in space, two primary factors demand comprehensive examination: space radiation and microgravity. This paper elucidates the multifaceted consequences of these stressors on astronaut health, emphasizing the imperative for innovative countermeasures to safeguard the physiological well-being of individuals engaged in long-duration space missions.

The space environment is replete with high-energy particles, originating from both solar and cosmic sources, collectively referred to as space radiation [28]. This radiation poses a significant threat to astronaut health, due to its ionizing nature which is capable of permeating spacecraft shielding and interacting with biological tissues [28]. Prolonged exposure to space radiation is known to induce DNA damage and compromise the immune system, rendering astronauts more susceptible to a spectrum of illnesses [29]. Robust countermeasures are requisite to mitigate the adverse physiological effects of space radiation, and ensure the sustained health of individuals navigating the extraterrestrial environment [30].

A defining characteristic of space habitation is the microgravity environment experienced within spacecrafts and space stations. The absence of gravitational cues challenges the homeostatic mechanisms of the human body, necessitating tailored interventions to counteract the deleterious effects of microgravity on astronaut health. Prolonged exposure to space radiation can damage DNA and weaken the immune system, making astronauts more susceptible to illnesses [31–34]. Finding effective countermeasures is crucial for successful long-duration space missions. The development of effective countermeasures assumes paramount significance in facilitating the adaptability of the human body to the challenges posed by microgravity during extended space missions.

Given this, the physiological consequences of staying in space, encapsulated by the dual stressors of space radiation and microgravity, underscore the necessity for meticulous examination and proactive mitigation strategies. Prolonged exposure to space radiation threatens the genetic integrity of astronauts [35] and compromises their immune resilience [36]. Concurrently, the microgravity environment induces skeletal and cardiovascular adaptations that necessitate targeted countermeasures. The pursuit of innovative solutions to ameliorate these physiological challenges is indispensable for the success of long-duration space missions, ensuring the sustained health and resilience of astronauts as they traverse the complexities of space exploration.

## 8. The Need for Biological Protection against Space Radiation

The occurrence of a large solar particle event (SPE) that leads to the exposure of astronauts to lethal doses of space radiation is considered to be a low probability event [37]. However, it is important to note that space radiation and its effects are still areas of active research, and our understanding of them continues to evolve. Solar particle events are caused by the release of highly energetic particles, primarily protons and occasionally heavier ions, from the Sun during solar flares or coronal mass ejections. These particles can pose a significant radiation hazard to astronauts, and can penetrate spacecraft shielding, thus potentially exposing astronauts to elevated radiation levels [37].

While SPEs can be detected and monitored, accurately predicting their occurrence and magnitude with a high degree of certainty is challenging. However, advances in space weather forecasting have improved our ability to provide early warnings of potential SPEs. Monitoring systems on spacecraft and satellites, as well as ground-based observatories, continuously observe the Sun and its activity in order to detect and track solar events that could lead to SPEs [38]. The magnitude of an SPE refers to the intensity of the particle flux, which is the number of particles per unit area per unit time. The magnitude of an SPE can vary significantly, ranging from minor events with relatively low particle fluxes to major events with high particle fluxes. Predicting the exact magnitude of an SPE is also challenging, as it depends on various factors, including the energy and direction of the particles emitted during the solar event and the characteristics of the interplanetary space through which these particles travel [39].

To mitigate the risks associated with SPEs, space agencies employ measures, such as active monitoring, early warning systems, and spacecraft shielding. Astronauts are also provided with dosimeters to monitor their radiation exposure during space missions. Additionally, mission planners aim to schedule extravehicular activities (EVAs) and critical operations during periods of lower solar activity to minimize the potential for high radiation exposure [40]. Therefore, while SPEs can pose a risk to astronauts in space, the space

agencies and researchers are actively working to improve our understanding, monitoring capabilities, and mitigation strategies to ensure astronaut safety.

## 9. Advancing Biological Protection Strategies

Recognizing that conventional physical shielding may prove insufficient, there is a need for a multifaceted approach to radiation protection. Specifically, it proposes the pre-flight screening of astronauts to assess their adaptive responses, investigations into methods for enhancing their immune system resilience, and deliberations on the potential use of radiation effect modulators. Deep space exploration necessitates a paradigm shift in radiation protection strategies for astronauts due to the heightened exposure to ionizing radiation, posing considerable health risks, including carcinogenesis, CNS damage, cardiovascular diseases, and the potential for ARS during SEP events. Conventional approaches relying solely on physical shielding fall short, prompting an exploration of innovative biological protection methods. This paper delves into the proposal of pre-flight screening to evaluate adaptive responses, investigations into immune system augmentation, and the potential use of radiation effect modulators. It particularly underscores the role of vitamin C as a promising radiation mitigator, acknowledging its non-toxic nature, cost-effectiveness, and accessibility, while acknowledging the need for thorough investigation in the context of human space exploration.

The complex composition of ionizing radiation in deep space, comprising neutrons, protons, and heavy ions from GCR, SW, and SEP, introduces unique challenges for astronaut health. Conventional risks include carcinogenesis, CNS damage, cardiovascular diseases, and the potential for ARS during significant SEP events. Physical shielding alone proves inadequate, necessitating a comprehensive biological protection approach to safeguard astronaut well-being during prolonged deep space missions. To address the multifaceted challenges posed by ionizing radiation, this paper proposes a three-pronged approach, encompassing pre-flight screening, immune system enhancement, and the exploration of radiation effect modulators. Pre-flight screening aims to evaluate aspirants for their adaptive responses to radiation, allowing for a tailored selection of astronauts better equipped to withstand its effects. Investigating methods to boost the immune system further enhances astronauts' resilience to radiation-induced health risks. Additionally, the examination of radiation effect modulators presents a promising avenue for intervention.

In summary, as the demands of deep space exploration necessitate innovative approaches to mitigate ionizing radiation risks, this paper advocates for a comprehensive biological protection strategy. The proposed pre-flight screening, immune system enhancement, and exploration of radiation effect modulators offer a multifaceted framework for advancing astronaut safety.

## 10. Protective Role of Vitamin C against Ionizing and Non-Ionizing Radiation

Over the past decade, Mortazavi et al. have conducted several studies on the protective role of vitamin C against ionizing and non-ionizing radiations. One study found that high doses of vitamin C can show life-saving radioprotective effects [41]. Another study investigated the potential radiation mitigation effect of vitamin C, and found that it can serve as an antioxidant to protect DNA damage caused by exposure to ionizing radiation [42]. They have also explored the possible applications of vitamin C in manned deep space missions, where it could be used as a radiation mitigator to protect astronauts from the harmful effects of radiation [42]. As illustrated in Figure 1, these studies suggest that vitamin C holds potential as a promising radioprotective agent and could therefore be useful in protecting against the harmful effects of radiation.

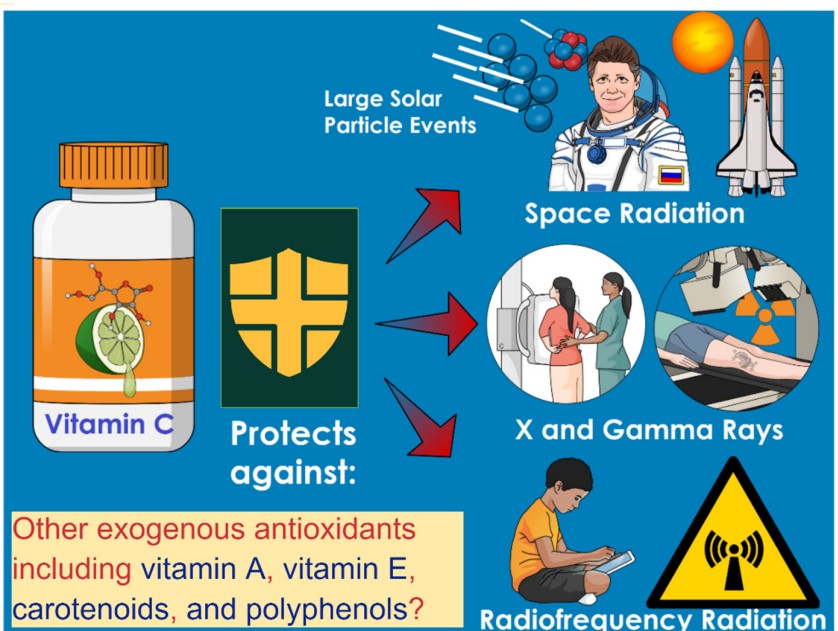

**Figure 1.** Protective role of vitamin C against both ionizing and non-ionizing radiations is confirmed in several studies. (Figure produced based on the references [5,42,43]).

## 11. Chemical Stability of Ascorbic Acid

Vitamin C is a sensitive compound that degrades when exposed to factors such as light, heat, metal ions ($Cu^{2+}$, $Fe^{2+}$, $Zn^{2+}$), and an alkaline pH. It can even denature at temperatures as low as 86 degrees Fahrenheit, making it vulnerable during cooking. In a dry state, it remains stable, but in a solution, it rapidly oxidizes. To address this challenge, researchers are developing methods to encapsulate vitamin C for protection. Techniques like microfluidic, melt extrusion, spray drying, and chilling have been employed, primarily producing microscale particles. In specific conditions, nanoscale encapsulation can be achieved through ion gelation of chitosan or complex coacervation with anionic polymers. To prevent the degradation of vitamin C and other antioxidants in space missions, it is important to consider the nutritional stability of these compounds under spaceflight conditions, as some vitamins and antioxidants may degrade during storage in space. In these cases, besides cultivating vitamin C-rich vegetables for astronauts to use fresh sources of vitamin C, designing a safe box with a high-level attenuation of space radiation would be of crucial importance. Low-molecular-weight bioactive compounds can also shield vitamin C by neutralizing factors that cause degradation in the solution [44].

## 12. Why Do We Need Fresh Sources of Vitamin C in Space?

When astronauts are in space, they tend to build up too many ROS (reactive oxygen species) because of the effects of microgravity and radiation. To counteract this, providing antioxidants could be beneficial in promoting astronaut well-being by reducing this occurrence [6]. An excessive increase in ROS levels may require the use of exogenous antioxidants like vitamin A, vitamin C, vitamin E, carotenoids, and polyphenols [45]. ROS can negatively impact a range of processes, including redox regulation, cell signaling, the promotion of proliferation, immunity, apoptosis, autophagy, and necrosis [46].

It is known that space radiation can have an impact on the structure and function of various drugs, including but not limited to epinephrine. Space radiation can cause the ionization and fragmentation of the molecules in drugs such as epinephrine, leading to changes in their chemical structure and potentially affecting their effectiveness. Some examples of other drugs that may be affected by space radiation include antibiotics, anticancer drugs, and painkillers [47]. The extent of the impact can vary depending on factors such as the specific drug and the duration and intensity of the exposure to radiation [48]. Given

this consideration, as scientists cannot fully protect drugs such as vitamin C from space radiation during long-term space missions to, for example, Mars or even beyond, access to fresh and rich sources of vitamin C would be crucial. Figure 2 shows how the accumulation of excess ROS in space, caused by microgravity and radiation, can potentially be reduced by administering antioxidants such as vitamin C.

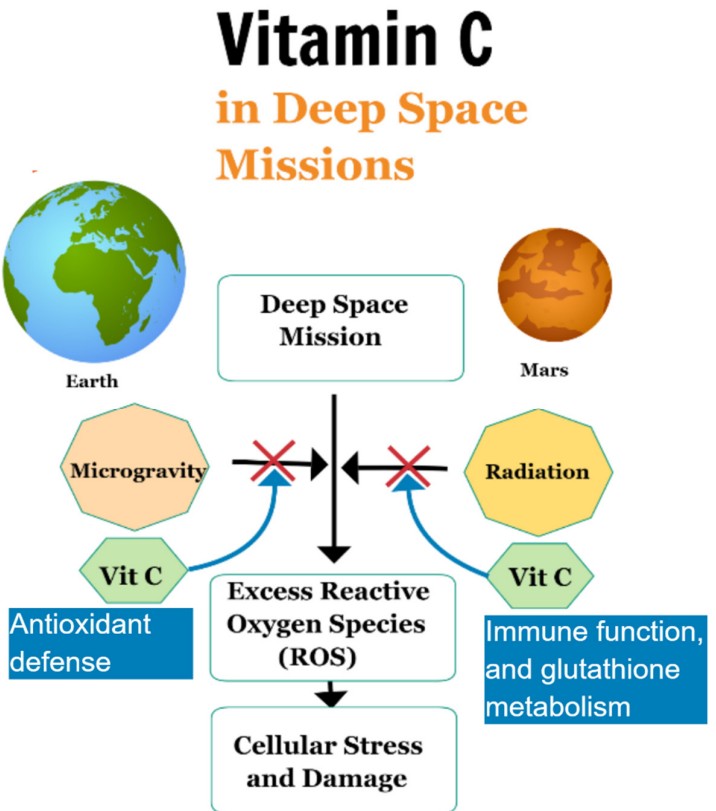

**Figure 2.** The accumulation of excess ROS in space24, caused by microgravity and radiation, can potentially be reduced by administering antioxidants such as vitamin C. This could potentially improve the health of astronauts (Figure produced based on the reference [6]).

## 13. The History of Microgreens for Cultivation in Space

While growing vegetables in space presents challenges, NASA has successfully tested growing a variety of vegetables aboard the ISS using Veggie [49]. Growing food in space has several benefits, including providing fresh food and variety in astronaut diets. NASA has been growing fresh vegetables on the International Space Station (ISS) using the Veggie system since 2014 [50]. Astronauts have grown eight different types of leafy greens, including red and green romaine lettuce, Chinese cabbage, mustard, and Russian kale [51]. While consuming vegetables high in vitamin C can provide several health benefits, including immune system support [52], there is limited research on whether it can help astronauts cope with the problems associated with space radiation and microgravity, the two key stressors in space environment. It is important to note that the vitamin C content of vegetables can vary depending on factors such as the variety, ripeness, and cooking method.

Astronauts aboard the International Space Station (ISS) have been successfully cultivating kale and other vegetables as part of their ongoing efforts to sustain themselves during long-duration space missions. This groundbreaking initiative aims to provide fresh and nutritious food for the crew, while also studying the potential for sustainable agriculture in space [53].

Inside the ISS, a designated area called the "Veggie Plant Growth System" has been set up to facilitate the growth of various plants. This system utilizes LED lights to provide the necessary light spectrum for photosynthesis, while a carefully controlled environment

ensures optimal temperature, humidity, and nutrient levels [54]. To explore the potential of various plant species that might be highly effective in combating radiation-induced damage under space conditions, we should focus on plants that have demonstrated radioprotective properties. In this context, plants such as *Podophyllum hexandrum*, *Amaranthus paniculatus*, *Emblica officinalis*, *Phyllanthus amarus*, *Piper longum*, *Tinospora cordifolia*, *Mentha arvensis*, *Mentha piperita*, *Syzygium cumini*, *Zingiber officinale*, *Ageratum conyzoides*, *Aegle marmelos*, and *Aphanamixis polystachya* have been shown to protect against radiation-induced lethality, lipid peroxidation, and DNA damage [8].

Kale, a leafy green vegetable rich in vitamins and minerals, has been a popular choice for cultivation due to its hardiness and nutritional value [55]. The astronauts carefully tend to the plants, monitoring their growth, and ensuring they receive the right amount of water and nutrients.

The benefits of cultivating vegetables in space are manifold. Firstly, it provides astronauts with a fresh food source, reducing their reliance on pre-packaged meals and increasing their overall well-being. Fresh produce not only improves the crew's morale, but also contributes to their physical health by providing essential nutrients [56].

Moreover, growing plants in space has broader implications for future space exploration and colonization. It helps scientists understand the potential for sustainable agriculture in environments with limited resources, such as on other planets or in space habitats. By studying how plants adapt and thrive in microgravity, researchers can develop innovative techniques to support long-duration missions and establish self-sustaining colonies in space [57].

The cultivation of kale and other vegetables inside the ISS represents a significant milestone in the quest for sustainable space exploration. It showcases the ingenuity and resourcefulness of astronauts and scientists working together to overcome the challenges of living and thriving in the harsh environment of space. As we continue to push the boundaries of human exploration, the ability to grow fresh food in space will undoubtedly play a crucial role in our journey to the stars.

## 14. Choosing the Best Microgreens for Cultivation in Space Is a Big Challenge

Microgreens are small, fast-growing crops that are valued for their color, flavor-enhancing properties, and rich phytonutrient content. A recent study has focused on the selection of microgreens for cultivation in space [58]. The researchers developed an algorithm to compare different genotypes of microgreens based on various parameters related to growth, phytonutrients, and mineral elements. The selection process consisted of two phases. In the first phase, the researchers used data from the literature to generate a ranking list of microgreens. This list was based on 25 parameters, including growth characteristics and the presence of phytonutrients such as tocopherol, phylloquinone, ascorbic acid, polyphenols, lutein, carotenoids, and violaxanthin. In the second phase, germination and cultivation tests were conducted on the top six species identified in the first ranking list. Based on the results, radish and savoy cabbage were ranked highest in terms of productivity and phytonutrient profile. The algorithm and selection method used in this study provide an objective way to compare and rank candidate species for cultivation in space. This approach can also be adapted for new species or specific selection purposes by modifying the parameters or prioritization criteria [58].

## 15. Why Diet Extremely Matters in Space

As humans venture farther into the cosmos, it becomes increasingly important to address the physiological challenges associated with extended space missions. Among these challenges, the impact of space radiation and microgravity on human health has emerged as a significant concern [59]. Nutritional balance is crucial for astronaut health and mission success, and nutraceuticals from various sources, such as whole grains, marine algae, edible mushrooms, phytochemicals, vitamins, and minerals, can play a role in protecting cells from radiation-induced injury and oxidative stress. Including these nutraceuticals in the

astronauts' regular diet can help maintain nutritional balance and prevent diseases caused by oxidative stress during long-term space mission.

Researchers and space agencies are exploring innovative strategies to mitigate these effects and improve astronauts' resilience. One such approach involves harnessing the power of artificial intelligence to guide the selection and cultivation of vitamin C-rich vegetables, thus offering a promising solution to bolstering astronaut health.

In the quest for extended space missions and interplanetary travel, ensuring the health and well-being of astronauts is paramount. One of the key challenges faced by astronauts in space is the detrimental impact of high levels of space radiation and microgravity on their overall health, particularly their immune systems [60]. To counteract these effects, researchers are turning to artificial intelligence (AI) to assist astronauts in selecting and cultivating vitamin C-rich vegetables. This innovative approach not only contributes to better resistance against the harsh space environment, but also enhances the nutritional quality of astronauts' diets.

## 16. The Role of Vitamin C in Astronaut Health

Vitamin C, renowned for its antioxidant properties and immune-boosting effects, plays a pivotal role in maintaining astronaut health. Adequate vitamin C intake can help mitigate the oxidative stress caused by space radiation and enhance the body's ability to repair DNA damage. Furthermore, vitamin C supports collagen production, which is critical for maintaining healthy skin, blood vessels, and connective tissues, which are susceptible to deterioration in microgravity conditions [6].

## 17. Rich Sources of Vitamin C

Radish [61] and savoy cabbage [62] are indeed good sources of vitamin C, but they may not be considered the richest fresh vegetables in terms of vitamin C content. While radishes and savoy cabbage contain moderate amounts of vitamin C, there are several other vegetables that are higher in this nutrient. For instance, bell peppers (particularly red and yellow varieties), kale, broccoli, Brussel sprouts, and cauliflower are generally recognized as vegetables with higher vitamin C content compared to radishes and savoy cabbage [63]. Citrus fruits like oranges and grapefruits are also well-known for their high vitamin C content [64]. It is worth noting that the vitamin C content can vary depending on certain factors, such as the vegetable's freshness, growing conditions [65], and storage methods [66]. To maximize the vitamin C content in vegetables, it is recommended to consume them when they are fresh and not overcook them, as vitamin C is sensitive to heat and can be lost during cooking [67].

## 18. Our Experience on Vitamin C as a Mitigator of Space Radiation

Vitamin C has been proposed as a potential radiation mitigator for astronauts exposed to space radiation. Studies have shown that vitamin C can decrease radiation-induced chromosomal damage in rodents, but further investigation is needed to determine its effectiveness in humans [3]. As illustrated in Figure 3, rats treated with a single dose of vitamin C up to 24 h after exposure to gamma radiation showed improved survival rates and cell viability [2]. Simulated microgravity, which is experienced during space-flight, has been found to increase heavy ion radiation-induced cell apoptosis in human B lymphoblasts. However, antioxidants such as N-acetyl cysteine (NAC) and quercetin have been shown to reverse this effect, suggesting their potential as protective agents for astronauts [68]. Developing an antioxidant cocktail to prevent or mitigate reactive oxygen species (ROS) damage during space exploration is also being considered as a strategy to maintain astronaut health [6].

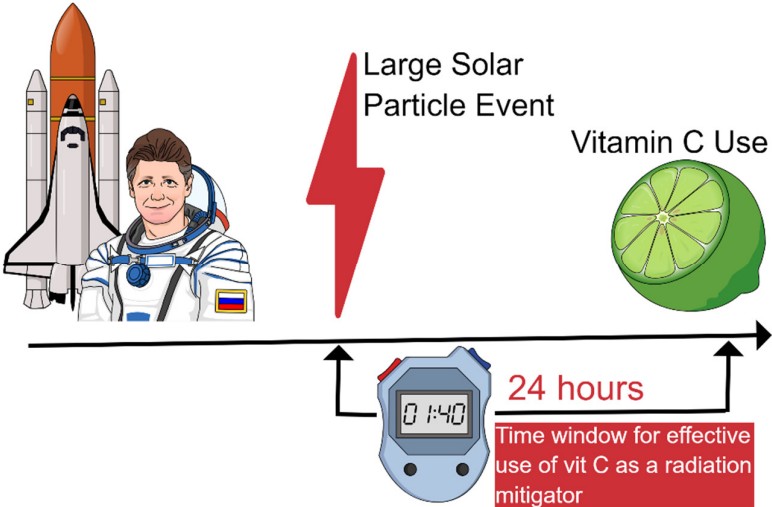

**Figure 3.** Rats treated with a single dose of vitamin C up to 24 h after exposure to gamma radiation showed improved survival rates and cell viability. The findings of this study can pave the road for potential mitigation of the effects of exposure to unpredictable large SPEs.

Based on this background, supplements that benefit regulating the somatic function have been approved to reduce the influence caused by cosmic radiation, microgravity, and hypo-magnetic environment in space [5]. Vitamin C is a prospective safe and available radiation mitigator when used several hours after exposure to radiation [2,69] Recently, vitamin C has also been reported to have a role as a radioprotector [2,70,71], thus, in a space mission context, can help neutralizing free radicals and protecting cells from damage induced by ionizing radiation [6].

## 19. Conclusions

Vitamin C emerges as a promising candidate for mitigating space radiation's effects, offering potential benefits, even when administered after exposure. The success of vegetable cultivation on the ISS indicates the feasibility of providing fresh, vitamin C-rich food for astronauts. However, challenges like the variable vitamin C content in vegetables and the need for stable formulations in space conditions persist. The protective role of vitamin C against both ionizing and non-ionizing radiations underscores its significance in safeguarding astronaut health. As researchers explore the potential of microgreens for space cultivation, selecting optimal varieties becomes critical. Integrating artificial intelligence in vegetable selection and cultivation may enhance astronaut diets and resilience. While the threat of solar particle events remains, ongoing advancements in monitoring and mitigation strategies demonstrate a commitment to astronaut safety. As we venture into extended space missions, understanding the crucial role of vitamin C and maintaining a balanced diet becomes imperative for astronaut health and well-being.

**Author Contributions:** Conceptualization, A.M. and L.S.; literature review, A.M., L.S. and D.F.; writing—original draft preparation, A.M.; writing—review and editing, A.M., H.Y., B.F.B.B.Z., R.Y., F.G.-M., S.M.J.M., M.H., D.F., L.S.; supervision, A.M. and L.S. All authors have read and agreed to the published version of the manuscript.

**Funding:** This research received no external funding.

**Acknowledgments:** The authors would like to thank the INIRPRC for their technical support.

**Conflicts of Interest:** The authors declare no conflicts of interest.

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
