# Peer review of "Cultivation of Vitamin C-Rich Vegetables for Space-Radiation Mitigation"

_radiation, doi:10.3390/radiation4010008_

Round 1

Reviewer 1 Report

Comments and Suggestions for Authors

The review, or rather study of an issue, presented by Dr. Mortazavi et al., argues and justifiably promotes vitamin C as a radioprotector of antioxidant nature. The manuscript is easy to read and well structured, but lacks a bit of scientific rigor. Relying on this, I believe that the manuscript can be accepted after a minor revision.

 Minor claim

1) The Introduction section lacks general information on currently available approaches to radio protection, including cases of exposure to heavy charged particles. Typical approaches are well covered, for example, in the following review https://doi.org/10.18632/oncotarget.24461, or atypical approaches, for example, in the following study: https://doi.org/10.2174/1871527320666210122092330. It would be sufficient to provide a brief overview and categorization of the use of vitamin C within the existing literature.

2) While the article includes three illustrations, they are not scientifically oriented, but rather serve popular science or promotional purposes. At least two of the figures should be removed or replaced with content relevant to the review's focus. It would be good to visualize a number of metabolic processes/molecular targets that vitamin C could potentially act on.

3) The figures are not numbered; all have been assigned the number 1.

Author Response

Please, see attachment.

Reviewer 2 Report

Comments and Suggestions for Authors

Mortazavi et al., in their manuscript titled "Cultivation of Vitamin C-Rich Vegetables for Space Radiation Mitigation", provide a very succinct and pertinent review of an aspect of an emerging field of research called Nutraceuticals. They  summarize the progress in cultivation of vitamin C-rich vegetables for the mitigation of space radiation. Among the many strengths of this review are:

1. Clearly itemized areas of review, 6 of them stated in the introduction.

2. Excellent figures, adapted and supplemented based on publications reviewed.

3. Clear statement of challenges, possible solutions and calls for new research.

My corrections/suggestions for improvement are:

1. All the Figures are labelled as Figure 1. Please correct these (Figure 2, Figure 3....

2. Connections with radiation therapy on Earth could be added to enhance scientific robustness of the review. In space, radiation is "harmful". On Earth, radiation is used to treat cancer. Interestingly, cancer patients receive Vitamin C alongside radiation. Is Vitamin C, a radiation mitigator in space medicine and a radiation enhancer (radiosensitizer) in terrestrial medicine. This reference (Khazaei S, Nilsson L, Adrian G, Tryggvadottir H, Konradsson E, Borgquist S, Isaksson K, Ceberg C, Jernström H. Impact of combining vitamin C with radiation therapy in human breast cancer: does it matter? Oncotarget. 2022 Feb 22;13:439-453. doi: 10.18632/oncotarget.28204. ), and others might help.

3. A few more outstanding citations and summaries could be done in each of the 6 areas adumbrated in the introduction. For instance, Kulkarni, S., Gandhi, D., Mehta, P.J. (2022). Nutraceuticals for Reducing Radiation Effects During Space Travel. In: Pathak, Y.V., Araújo dos Santos, M., Zea, L. (eds) Handbook of Space Pharmaceuticals. Springer, Cham. https://doi.org/10.1007/978-3-030-05526-4_54, is almost a must, for this review.

Comments on the Quality of English Language

Minor English language editing.

Author Response

Please, see attachment.

Reviewer 3 Report

Comments and Suggestions for Authors

After reviewing the article "Cultivation of Vitamin C-Rich Vegetables for Space Radiation Mitigation," I have identified several areas for improvement along with reasons to include the suggested references. Here is the feedback:

Negative Feedback and Areas for Improvement:

  1. Lack of Specificity in Microgravity Research: The paper broadly discusses the challenges of space radiation and microgravity but lacks specific focus on how microgravity affects the growth and nutritional quality of plants like wheatgrass, which are rich in antioxidants such as vitamin C.

  2. Insufficient Exploration of Varied Plant Species: While the paper emphasizes the importance of cultivating vitamin C-rich vegetables, it doesn't thoroughly explore the potential of various plant species, especially those that might be more effective in combating radiation-induced damage under space conditions.

  3. Limited Discussion on Antioxidant Properties Beyond Vitamin C: The paper primarily focuses on vitamin C and its benefits. However, it does not adequately address the broader spectrum of antioxidant properties that other compounds in plants might offer, which are crucial in mitigating radiation effects.

  4. Overemphasis on Vitamin C Stability: There is considerable emphasis on the stability and degradation of vitamin C in space conditions, but there is a lack of practical solutions or alternative approaches to ensure its efficacy in space missions.

  5. Methodological Limitations: The article could benefit from more detailed methodological descriptions, particularly regarding the cultivation techniques and challenges of growing vegetables in space environments.

Recommendations to Include Suggested Reference:

  1. Al-Awaida W, Al-Ameer HJ, Sharab A, Akasheh RT. (2023): This study can provide valuable insights into how simulated microgravity conditions influence the properties of wheatgrass, a potential source of antioxidants like vitamin C. Its inclusion would add depth to the paper's discussion on cultivating plants in space environments.

Conclusion:

The paper presents a relevant topic but needs to address the outlined limitations to enhance its scientific rigor and scope. Incorporating the suggested references will not only strengthen the current discussion but also expand the paper's perspective on plant cultivation under microgravity conditions and their broader health benefits in space exploration scenarios.

Author Response

Please, see attachment.

Round 2

Reviewer 3 Report

Comments and Suggestions for Authors All comments have been adressed